# Amino Acid-Assisted Sand-Milling Exfoliation of Boron Nitride Nanosheets for High Thermally Conductive Thermoplastic Polyurethane Composites

**DOI:** 10.3390/polym14214674

**Published:** 2022-11-02

**Authors:** Shihao Zheng, Bing Wang, Xiaojie Zhang, Xiongwei Qu

**Affiliations:** Hebei Key Laboratory of Functional Polymers, Department of Polymer Materials and Engineering, Hebei University of Technology, 8 Guangrong Street, Tianjin 300130, China

**Keywords:** boron nitride nanosheets, sand-milling exfoliation, thermoplastic polyurethane, thermally conductive material, amino acid

## Abstract

Boron nitride nanosheets (BNNSs) show excellent thermal, electrical, optical, and mechanical properties. They are often used as fillers in polymers to prepare thermally conductive composites, which are used in the production of materials for thermal management, such as electronic packaging. Aside from the van der Waals force, there are some ionic bond forces between hexagonal boron nitride (h-BN) layers that result in high energy consumption and make BNNSs easily agglomerate. To overcome this issue, _L_-lysine (Lys) was first employed as a stripping assistant for preparing graft-functionalized BNNSs via mechanical sand-milling technology, and the obtained Lys@BNNSs can be added into thermoplastic polyurethane (TPU) by solution mixing and hot-pressing methods to prepare thermally conductive composites. This green and scalable method of amino acid-assisted sand-milling can not only exfoliate the bulk h-BN successfully into few-layer BNNSs but also graft Lys onto the surface or edges of BNNSs through Lewis acid–base interaction. Furthermore, benefiting from Lys’s highly reactive groups and biocompatibility, the compatibility between functionalized BNNSs and the TPU matrix is significantly enhanced, and the thermal conductivity and mechanical properties of the composite are remarkably increased. When the load of Lys@BNNSs is 3 wt%, the thermal conductivity and tensile strength of the obtained composites are 90% and 16% higher than those of the pure TPU, respectively. With better thermal and mechanical properties, Lys@BNNS/TPU composites can be used as a kind of heat dissipation material and have potential applications in the field of thermal management materials.

## 1. Introduction

The transition from multi-layered stacked three-dimensional (3D) materials to two-dimensional (2D) nanosheets is accompanied by significant enhancements in various physical and chemical properties. Since the exfoliation of boron nitride nanosheets (BNNSs) was first accomplished in 2005 [1], their excellent mechanical, optical, thermal, and electrical properties have attracted worldwide attention. The special arrangement of boron atoms and nitrogen atoms leads to the boron nitride layers being bound together by van der Waals forces and partially by ionic forces [2]. Therefore, the method of large-scale exfoliation of hexagonal boron nitride (h-BN) is suitable for producing 2D crystals with high quality and high purity. Many feasible boron nitride exfoliation methods have been proposed through the unremitting efforts of researchers, such as ball-milling exfoliation, liquid phase exfoliation, ion intercalation exfoliation, and solvothermal exfoliation [3,4,5,6]. The discovery of many exfoliation methods has enabled us to obtain nanoparticles of different shapes and sizes, giving them high surface area, better chemical stability, high thermal conductivity, excellent high-temperature oxidation resistance, and electronic insulation performance. Therefore, BNNSs have broad application prospects in polymer matrix composites [7,8], electrocatalysis [9,10], adsorbents [11,12], hydrogen storage materials [13,14], and other fields.

With remarkable physical and chemical properties, BNNSs are often used as fillers to prepare thermally conductive composites in the polymer matrix, which has application potential in the field of thermal management materials [15,16,17]. The team of Mikhael Bechelany and Philippe Miele has developed a green, simple, and effective method for the exfoliation of BNNSs using water as a solvent and biomass gelatin as an exfoliation agent. The method enables not only effective BNNS exfoliation but also homogeneous dispersion of graphene-like BNNSs in gelatin solutions, which significantly improves the gas barrier properties of gelatin films [18]. Subsequently, this team introduced chloride ion intercalation on this basis, using a combination of ultrasound and centrifugation, which greatly improved the yield of BNNSs [19]. Due to the lack of reactive groups on the BNNS surface, surface functionalization has been carried out to reduce agglomeration and enhance the interaction with the polymer matrix. In addition, the mixing method between filler and polymer matrix is also a key element for the successful preparation of thermally conductive composites [20]. The common methods mainly include solution mixing, melt blending, in situ polymerization, and ball milling. Solution mixing is the traditional method for the production of h-BN/polymer composites, and the content of filler in the polymer matrix can be controlled relatively accurately through this method [20,21,22].

The surface of bulk h-BN generally shows low reactivity and is difficult to covalently modify. In the surface-functionalization-assisted exfoliation method, noncovalent modification approaches based on the Lewis acid-base reaction mechanism aroused the interest of researchers. Recently, natural amino acids, a kind of bio-renewable raw material, have attracted considerable interest because they are extensive sources of rich reactive functional groups and have excellent biocompatibility and biodegradability [23,24,25]. The molecular architecture of _L_-lysine shows that it has highly reactive amino and carboxyl groups as well as R-side chains. During the sand-milling process, the shear force between h-BN, exfoliation solvents, and zirconium dioxide (ZrO_2_) balls helps h-BN expose more electron-deficient B atoms which can interact with amino groups in Lys while enabling the stripping and functionalization of BNNSs. 

In this study, Lys and h-BN were dispersed uniformly in water by sand-milling methods. It was hoped that the Lys could be grafted onto BNNSs by the shear force and compression force developed during the sand-milling process to obtain functional boron nitride nanosheets (Lys@BNNSs). The effects of functionalized BNNS content (0.1 to 5 wt%) on the thermal and mechanical properties of composites were investigated. A series of composites were produced via solution mixing and hot pressing using Lys@BNNSs and h-BN as fillers, respectively. The thermal conductivity and mechanical properties of TPU composites were analyzed and discussed. Our work provides meaningful instructions on the optimum solution of green and scalable exfoliation of h-BN and is validated in high-performance TPU composites.

## 2. Materials and Methods

### 2.1. Materials

h-BN (lateral dimension 5 μm, industrial purity >99%, CAS: 10043-11-5) was provided by Shandong Qingzhou Materials Technology Co., Ltd., Shandong, China. _L_-Lysine (Lys, AR, CAS: 56-87-1) was purchased from Saen Chemical Technology Co., Ltd., Shanghai, China. TPU (1180A10) was procured from BASF China Co. (BASF, Shanghai, China). N,N-Dimethyl formamide (DMF, AR, CAS: 68-12-2) was provided by Tianjin Concord Technology Co., Ltd., Tianjin, China. Anhydrous ethanol (EtOH, AR, CAS: 64-17-5) was supplied by Tianjin Chemical Reagent Factory No. 6., Tianjin, China.

### 2.2. Preparation of Lys@BNNSs

The Lys@BNNSs were prepared by means of a sanding-assisted stripping method, the entire sanding process being carried out by a dispersion-mixing sanding multi-purpose machine (XD-B11OO, Shanghai Mu Xuan Industrial Co., Ltd., Shanghai, China). Firstly, 2 g h-BN powder was added to a quantity of aqueous lysine solution with a mass fraction of 2 wt% and dispersed well. The mixed dispersion and 200 g of ZrO_2_ balls (2 mm) were then poured into a stainless-steel sanding jar and sanded continuously at 2000 rpm for 8 h. When the sanding was completed, the ZrO_2_ balls were sieved out and the suspension was collected. The suspension was ultrasonically dispersed for 2 h and then centrifuged at 3000 rpm for 10 min to obtain the supernatant. Finally, the supernatant was centrifuged at 10000 rpm for 20 min, and the product was collected and dried at 80 °C to obtain Lys@BNNSs.

### 2.3. Preparation of TPU Composites

A series of TPU composites with different filler types and contents (0.1, 0.5, 1, 3, 5 wt%) were prepared by solution mixing and hot pressing. Firstly, Lys@BNNSs with different loadings were dispersed in 100 DMF to obtain a mixed dispersion, and then 10 g of TPU and the dispersion were poured into a round-bottom flask and mechanically stirred in an oil bath at 80 °C for 8 h to obtain a viscous liquid, which was poured into 1000 mL of ethanol to remove the DMF, filtered, and dried under vacuum at 50 °C for one day to obtain the product. The product was placed in a mold and heated at 180 °C under atmospheric pressure for 8 min, and then it was pressurized to 10 MPa for 10 min and cooled to room temperature to obtain Lys@BNNS/TPU composites. h-BN/TPU composites were prepared in the same way. The preparation process of h-BN/TPU composites is shown in Figure 1.

### 2.4. Measurements and Characterization

Chemical bonds and special groups of Lys, h-BN, and Lys@BNNSs were characterized by Fourier transform infrared spectroscopy (FTIR, Nicolet NEXUS 470, Thermo Fisher Scientific, Waltham, MA, USA). The thermal weight loss of Lys@BNNSs compared with h-BN was investigated by thermal gravimetric analysis (TGA, SDT/Q600, TA Instruments, New Castle, DE, USA) at a heating rate of 10 °C/min in a nitrogen atmosphere. The crystalline structure of h-BN and Lys@BNNSs was tested using X-ray diffraction (XRD, D8 Advance, Bruker AXS, Bruker AXS, Karlsruhe, Germany) at scan steps of 6°·min^-1^ from 10° to 80°. The elemental composition of Lys@BNNSs was analyzed by X-ray photoelectron spectroscopy (XPS, ESCALAB 250Xi, Thermo Fisher Scientific, Waltham, MA, USA). The morphology and microstructure of h-BN, Lys@BNNSs, and TPU composites were evaluated using a scanning electron microscope (SEM, TESCAN GAIA3, TESCAN ORSAY HOLDING, a.s., Kohoutovice, Czech Republic) and transmission electron microscope (TEM, Talos F200S, FEI USA, Hillsboro, OR, USA). Dynamic mechanical properties were determined using a dynamic mechanical analyzer (DMA, Tritec 2000, Triton Technology Ltd., Hamilton, Bermuda, UK). The temperature rise rate was 3 °C/min, from −80 to 40 °C, for testing.

The thermal conductivity (TC) of composites was tested according to the TC 3000 measuring instrument of Xi’an Xiaxi Electronic Technology Co., Ltd., Xi’an, China. The test method is the transient hot wire method, which is to place the line heat source plate between two smooth and flat samples and impose appropriate pressure weight to make them contact closely. When the sample heat balance monitoring reaches the standard, a constant voltage current is used to heat the line heat source plate to transfer heat to the sample. The thermal transfer rate is based on the thermal conductivity of the test material. The thermal conductivity of the sample (λ) can be calculated by the following formula:
(1)$$\lambda =\frac{q}{4\pi (d∆T/\mathrm{dlnt})}$$
where q is the heat generated by the hot wire per unit time and unit length, ΔT is the temperature change of the hot wire, and t is the time of measurement.

The mechanical properties of TPU composites were measured with a stretching rate of 100 mm min^−1^ using a universal tensile tester (CMT6104, New Think Measurement Technology Co. LTD, Shenzhen, China). A thermal imaging camera (DS-2TPH10-3AUF, Hikvision Digital Technology Co., Ltd., Hangzhou, China) was used to register the surface temperature changes of composite materials during cooling.

## 3. Results and Discussion

### 3.1. Characterizations of Lys@BNNSs

In this study, functioned BNNSs have been successfully prepared via amino acid-assisted sand-milling exfoliation. To identify the chemical bonds, elemental composition, and modification degree of Lys@BNNSs, the production was further characterized by FTIR, TGA, and XPS. Infrared spectra of h-BN, _L_-lysine, and Lys@BNNSs as shown in Figure 2a, where h-BN and Lys@BNNSs both have absorption peaks near 1380 and 800 cm^−1^, corresponding to the in-plane stretching vibration peak and out-of-plane bending vibration peak of BN [26], respectively. In addition to the characteristic absorption peaks of h-BN, some new peaks appear in the infrared spectrum of Lys@BNNSs. The broad absorption peaks in the range of 3300–3600 cm^−1^ belong to the stretching vibration peaks of –NH_2_ and –OH; those at 2942 and 2831 cm^−1^ are the asymmetric stretching vibration peaks and symmetric stretching vibration peaks of C–H (–CH_2_), respectively; and the peak at 1583 cm^−1^ could be considered as the stretching vibration absorption peak of C=O [27]. The new absorption peaks appearing in the spectrum preliminarily indicate that the Lys molecules were grafted onto the BNNS particles. Thermal gravimetric analysis (TGA) was used to measure the modified amount of _L_-lysine on the surface of h-BN. Figure 2b shows the TGA curves of h-BN and Lys@BNNSs at a heating rate of 10 °C/min from room temperature to 800 °C in a nitrogen atmosphere. h-BN is thermally stable in the temperature range, while Lys@BNNS has two major weight loss regions at 131–206 °C and 206–574 °C, respectively, which are attributed to the decomposition of _L_-lysine on BN nanoparticles. In addition, the weight loss measured during thermal decomposition shows that about 5.2% of _L_-lysine was grafted on h-BN nanoparticles, caused by Lewis acid–base interaction. In Figure 2b, it can be seen that there are two obvious large peaks in the DTG curve between 100 and 300 °C, indicating that Lys@BNNS has been pyrolyzed twice in this temperature range. The two largest thermal weight loss rates are 143 and 186 °C, respectively. The weight loss in the first stage is due to the decarboxylation reaction of lysine, and the weight loss in the second stage is due to the pyrolysis of lysine primary pyrolysis products and some lysine molecules. Meanwhile, the report of XPS is shown in Figure 2d. There are four obvious characteristic peaks at 531.7, 397.4, 284.8, and 190.5 eV in the full survey spectra of Lys@BNNSs, which correspond to O1s, N1s, C1s, and B1s, respectively, and the tested values are in good agreement with reported values in previous literature [28,29]. Notably, between the peaks at 397.7 eV (C–N) and 398.0 eV (B–N), a new peak at 398.4 eV is observed from the N1s spectrum of Lys@BNNSs, which corresponds to B···N interactions between B atoms in boron nitride and amino groups in _L_-lysine [23]. In addition, the B1s spectrum of Lys@BNNSs can be divided into two different boron bonds with binding energies of 191.1 eV and 189.3 eV, corresponding to the B–N and B···O bonds; the B···O bonds may be introduced by the ultrasonic process caused by the hydroxyl group [30]. The degree of functionalization of BNNS is derived from the ratio of nitrogen atoms to carbon atoms. From the XPS results, it can be found that the ratio of elemental nitrogen to oxygen in h-BN is 11.88, and this ratio decreases to 4.23 with increasing functionalization. At the same time, the elemental C content increases by 5.48%, a result that is consistent with the results of the TGA test. The above results further proved that _L_-lysine was successfully grafted on boron nitride nanosheets.

XRD testing was carried out to characterize the crystal structure of Lys@BNNSs after sand milling. As shown in Figure 2c, h-BN and Lys@BNNSs both show the typical hexagonal structure of boron nitride. The characteristic diffraction peak intensity of Lys@BNNSs on the (002) plane is significantly lower than that of h-BN. In addition, it can be observed from the inset that the FWHM is significantly broadened. This is due to the high-speed grinding beads colliding with the material, and the generated normal impact force as well as tangential shear force overcome the interlayer force of h-BN and reduce the orderly arrangement and crystallinity of the structure [31]. This result further illustrates that BN was successfully exfoliated.

To investigate the micromorphology of h-BN and Lys@BNNSs, the samples were observed using an SEM. Figure 3a–d are SEM images of h-BN and Lys@BNNSs with different magnifications. It can be seen from Figure 3a,b that the pristine h-BN material with a lateral dimension of about 5 μm is loosely distributed on the conductive tape with smooth surfaces and edges. Compared with the original h-BN, the smooth texture of Lys@BNNSs after _L_-lysine-assisted sand-milling has disappeared, and both the size and thickness are significantly reduced compared with those of h-BN, as shown in Figure 3d, which shows obvious slip peel traces and split-layer structure caused by horizontal slip [32]. In addition, the slip-exfoliated BNNSs have good lateral dimensions and structural integrity, which is more evident in the high-magnification SEM images of Lys@BNNSs (Figure 3d). The results fully show that the sanding peeling method and the addition of Lys have an obvious exfoliation effect on h-BN.

Lys@BNNSs were further analyzed using a TEM for more detailed structural information. As shown in Figure 4a,b, Lys@BNNSs showed lower contrast under the transmission electron microscope, indicating a reduction in the number of BNNS layers, and the split-layer structure caused by horizontal slip can be observed, which echoes the above SEM test results. The inset in Figure 4b is the six-fold symmetric lattice structure of a typical h-BN single crystal [33], indicating that the exfoliated BNNSs are characterized by low defects and high crystallinity. Under further magnification, as shown in Figure 4c, the material is found to have an obvious lattice structure, where the distance between two adjacent points is 0.25 nm, corresponding to the h-BN (002) crystal plane distance between B-B or N-N atoms [34].

While characterizing the microscopic morphology of the sample, the element distribution analysis of Lys@BNNSs was carried out in this study. As shown in Figure 4d, Lys@BNNSs not only contain B and N elements belonging to boron nitride but also contain newly added C and O elements in Lys. The above results further proved that h-BN was successfully exfoliated and Lys was successfully grafted on BNNSs.

### 3.2. Characterizations of TPU Composites

Filled thermally conductive composites were prepared with h-BN and Lys@BNNSs as the thermally conductive phase and TPU as the matrix. Figure 5a shows the relationship between the thermal conductivity of TPU-based composites and the filler content. It can be seen that the thermal conductivity (TC) of composite material is dependent on filler content within a certain range, and the thermal conductivity of composite material shows a trend of monotonically increasing with the addition of filler. This is due to BNNSs beginning to contact each other to form a local heat conduction chain with the increase in filler content. The contact area between the fillers increases with the further increase in the content, which is more conducive to the formation of a continuous thermal conductivity network in the polymer matrix. In addition, the lamellar fillers in contact with each other reduce the interfacial thermal resistance, and more heat can be transferred along the thermally conductive network paths [35]. The results show that the λ of pure TPU is only 0.20 ± 0.006 W m^−1^ K^−1^, and the λ of the TPU composite is 0.49 ± 0.001 W m^−1^ K^−1^ when filled with 5 wt% h-BN, which is 125% higher than that of pure TPU. When TPU is filled with 5 wt% Lys@BNNSs, the λ of composites can reach 0.52 ± 0.006 W m^−1^ K^−1^, 160% higher than that of pure TPU. The λ of Lys@BNNS/TPU composites is always higher than that of h-BN/TPU composites; this is attributed to the presence of Lys improving the interface affinity between Lys@BNNSs and TPU. Lys molecules are like a bridge; one amino terminus is attached to h-BN, and the other amino terminus can form a hydrogen bond with –NH– in TPU. This interaction can effectively reduce the self-agglomeration of Lys@BNNS particles and increase their chemical crosslinking points in TPU, reduce phonon scattering, and improve the interface heat transfer efficiency, thus achieving higher thermal conductivity [36]. To demonstrate the advantages of the Lys@BNNSs prepared in this work for TC of reinforced composites, we also summarize previous work on TC of BN/polymer composites (Table 1). It can be seen that the present Lys@BNNS/TPU composites exhibit a high TC at low filler content compared to the reported BN polymer composites.

The effect of fillers on the mechanical properties of TPU composites was tested by tensile testing. As shown in Figure 5b, the tensile strength of pure TPU is 18.8 ± 0.5 MPa. The maximum value of h-BN/TPU was 23.2 ± 0.6 MPa when the addition amount of h-BN was 0.1 wt%, which was 23.4% higher than that of pure TPU. When 1 wt% Lys@BNNSs is added to Lys@BNNS/TPU, the tensile strength reaches the maximum value of 27.8 ± 0.7 MPa, which is 48.0% higher than that of pure TPU. The above results prove that the addition of an appropriate amount of filler can effectively improve the mechanical properties of the composites. Lys@BNNSs mainly achieve the purpose of strengthening TPU composites by resisting stress with high strength and modulus and realizing the transfer of stress by using the plastic flow of TPU and the interface adhesion between TPU and Lys@BNNSs. In addition, with the increase in Lys@BNNS content, more loading must be applied to improve the tensile strength of TPU when Lys@BNNSs are fractured. The mechanical properties of TPU composites are closely related to the interface structure of the two phases. Therefore, when the filler content of h-BN is added, the interface adhesion between h-BN and TPU is weak and the stress transfer efficiency is reduced due to its smooth base plane and almost no active sites. As a result, the tensile strength of h-BN/TPU composites is significantly reduced compared with Lys@BNNS/TPU [37]. The above phenomenon was further observed using an SEM. The results showed that the cross sections of pure TPU were flat and smooth (Figure 5d), while the cross sections of h-BN /TPU and Lys@BNNS/TPU with filler content of 1 wt% are rough (Figure 5e,f), which was caused by filler distributed at the interface. By contrast, the dispersion and order degree of filler in Lys@BNNS/TPU is much higher than that in h-BN/TPU, which also proves that the stress transfer efficiency of the former is higher than that of the latter, which is consistent with the above tensile strength test results. Figure 5c shows the elongation at break of h-BN/TPU and Lys@BNNS/TPU composites as a function of filler content. Compared with pure TPU, the elongation at break of the composites decreased to varying degrees with the increase in filler content. This is because Lys@BNNSs or h-BN acts as a rigid particle filler during the stretching process, which will increase the brittleness of the composite and limit the movement of the matrix polymer chain, causing it to fracture [38].

To further investigate the mechanical properties of pure TPU and Lys@BNNS/TPU composites under alternating stress, the dynamic mechanical properties of these samples were tested to determine their storage modulus (E′) and loss factor (tan δ). The storage modulus reflects the rigidity of the material, and the temperature corresponding to the maximum peak value of the loss factor curve is the glass transition temperature (Tg) of TPU composites. As shown in Figure 6, the energy storage modulus of pure TPU is about 2598 MPa, and the energy storage modulus of the nanocomposite is higher than that of pure TPU, increasing by 36% to 3542 MPa at 1 wt% and 49% to 3859 MPa at 5 wt%. This is because the stiffness of Lys@BNNSs dispersed in the polymer is very large, and the density of the hydrogen bond between Lys@BNNSs and TPU increases with the increase in Lys@BNNS content, which weakens the mobility of the TPU molecular chain, which leads to an increase in the E′ of the TPU composites [39]. Figure 6b shows the relationship between the loss factor and temperature. It can be seen from the curve that the glass transition temperature of the composite material moves to the high-temperature direction after the addition of filler and increases with the increase in the content. The glass transition temperature of 5 wt% Lys@BNNS/TPU is −29.8 °C, which is 9.1 °C higher than that of pure TPU. This indicates that the increasing hydrogen bond density between Lys@BNNSs and TPU severely limits the movement of the polyurethane molecular chain [40], which requires more energy for the transition between the glassy state and the high elastic state of the polymer.

To characterize the potential of h-BN/TPU and Lys@BNNS composites for thermal management applications, a thermal imager was used to record the change in sample surface temperature over time at the same initial temperature. Before the test, samples of the same size were placed in an oven at 80 °C for 6 h to ensure uniform heating; then, the samples were transferred to an insulated foam box at room temperature, and the surface temperature change was recorded with a thermal imager. As shown in Figure 7, compared with h-BN/TPU, the Lys@BNNS/TPU composite shows a significant decreasing trend of temperature when the temperature reaches 100 s. In addition, the steady-state temperature of h-BN/TPU tends to be stable at about 550 s, until the temperature no longer changes at 600 s, while the temperature of Lys@BNNS/TPU composite decreases faster overall, and the steady-state temperature is at about 450 s. It tends to be stable until the temperature does not change significantly at 500 s. This phenomenon can be attributed to the fact that BNNSs not only have a better diameter-to-thickness ratio than h-BN but also have better compatibility with TPU due to the _L_-lysine coating on the particles. Therefore, a more effective thermal conductivity network can be constructed with the same content added.

## 4. Conclusions

In summary, Lys-graft-functionalized BNNSs were prepared by mechanical sand-milling, and then the Lys@BNNS/TPU composites were prepared by solution blending and hot pressing. The Lewis acid–base interaction between h-BN and Lys promotes the exfoliation and functionalization of BNNSs simultaneously. Lys@BNNSs maintained the maximum structural integrity while obtaining thinner thickness and larger lateral dimensions, and more importantly, the successful grafting of Lys. The modifier Lys acts as a bridge to organically combine the inorganic filler BNNSs and TPU matrix, significantly improving the compatibility and hydrophilia of BNNSs. The addition of Lys enhances the interfacial interaction between Lys@BNNSs and TPU and effectively reduces phonon scattering and interface thermal resistance, resulting in better thermal conductivity and mechanical properties of the composites. When the Lys@BNNS content reaches 3 wt%, the thermal conductivity and tensile strength of the obtained composites are 90% and 16% higher than those of pure TPU, respectively. Lys@BNNS/TPU with better thermal and mechanical properties can be used as a heat dissipation material and has application potential in the field of thermal management materials.

## Figures and Tables

**Figure 1 polymers-14-04674-f001:**
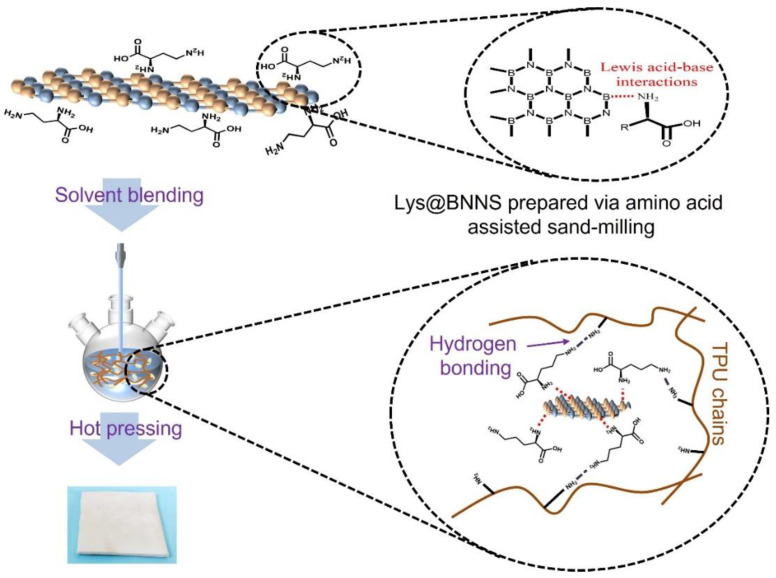
The preparation process of TPU composite materials.

**Figure 2 polymers-14-04674-f002:**
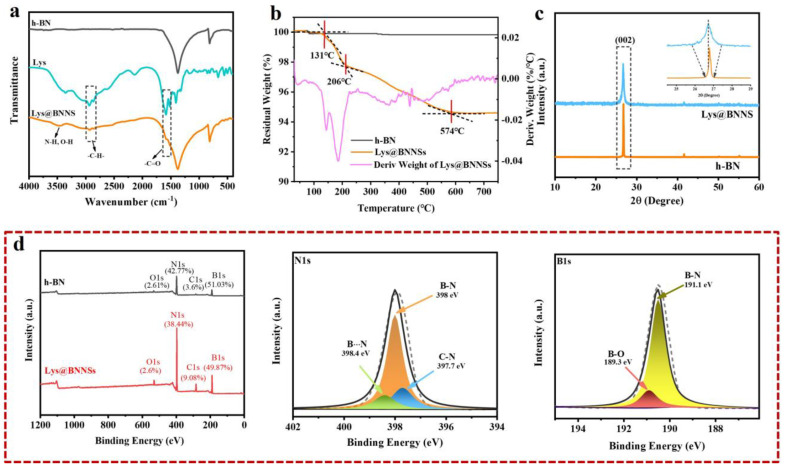
(**a**) FT-IR spectra of h-BN, Lys, and Lys@BNNSs. (**b**) TGA curves of h−BN and Lys@BNNSs and DTG curves of Lys@BNNSs. (**c**) XRD patterns of h-BN and Lys@BNNSs. (**d**) High-magnification XPS survey spectra of h-BN and Lys@BNNSs, XPS survey spectra N1s and B1s of Lys@BNNSs, and corresponding peak-fitting curves.

**Figure 3 polymers-14-04674-f003:**
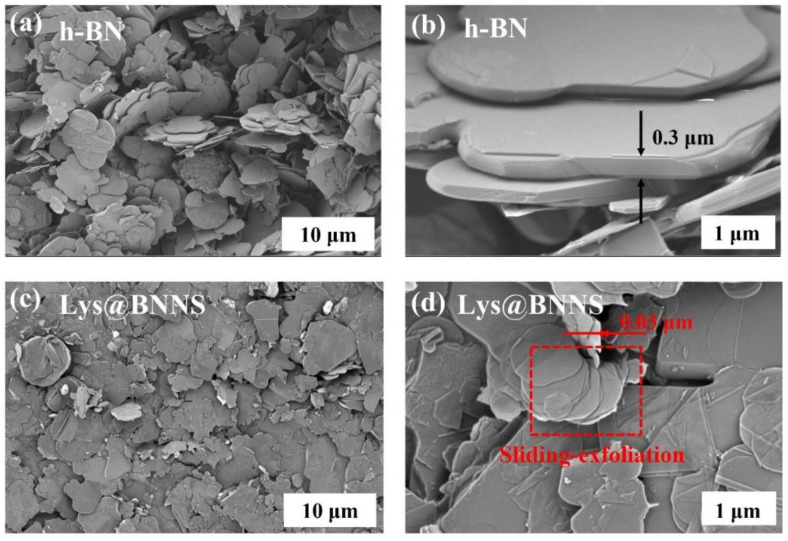
SEM images of (**a**,**b**) h-BN and (**c**,**d**) Lys@BNNSs.

**Figure 4 polymers-14-04674-f004:**
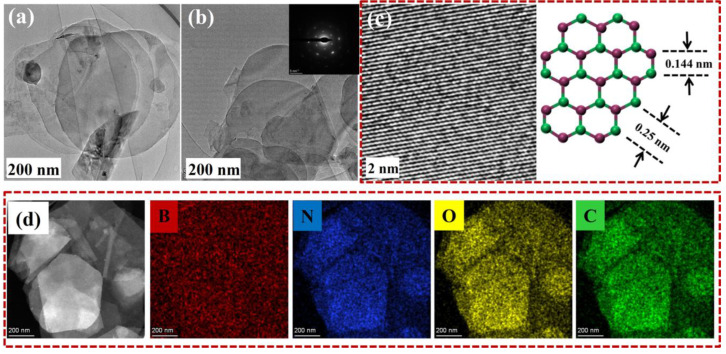
(**a**,**b**) TEM image of the Lys@BNNSs, with the inset of (**b**) showing the selected area electron diffraction pattern illustrating typical six-fold symmetry; (**c**) HRTEM image of Lys-BNNSs and a representative crystal structure of the basal plane of h-BN; (**d**) HR elemental mapping of Lys@BNNSs.

**Figure 5 polymers-14-04674-f005:**
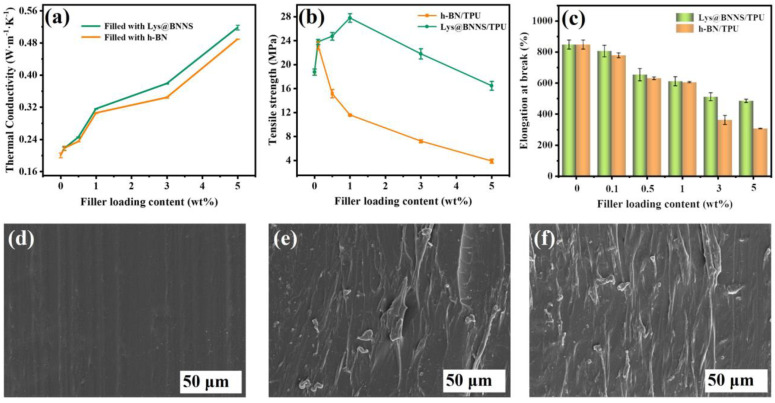
(**a**) Thermal conductivity, (**b**) tensile strength, and (**c**) elongation at break of composites as a function of filler content. SEM images of (**d**) TPU, (**e**) 0.1 wt% h-BN/TPU, and (**f**) 0.1 wt% Lys@BNNS/TPU.

**Figure 6 polymers-14-04674-f006:**
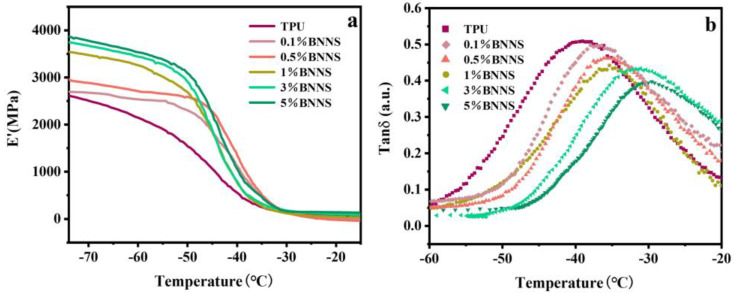
(**a**) Dynamic mechanical analysis storage modulus (E′) and (**b**) loss factor (tan δ) curves of TPU composites.

**Figure 7 polymers-14-04674-f007:**
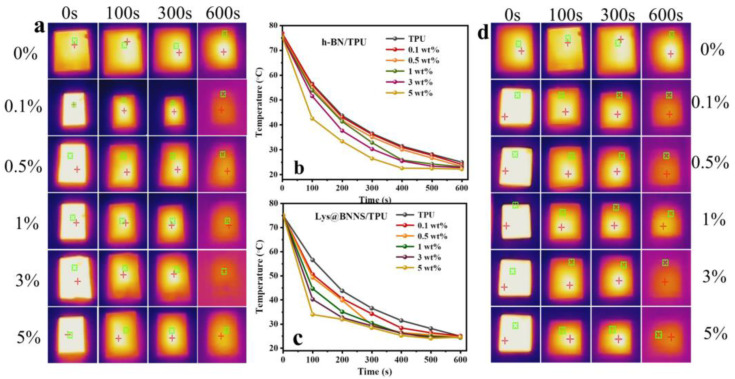
(**a**) Infrared thermal images and (**b**) surface temperature variation with cooling time for TPU and h-BN/TPU; (**c**) infrared thermal images and (**d**) surface temperature variation with cooling time for TPU and Lys@BNNS/TPU.

**Table 1 polymers-14-04674-t001:** TC of our Lys@BNNS/TPU composites in this study and BN/polymer composites which were previously reported.

Filler	Matrix	Filler Content	TC (W·m^−1^·K^−1^)	Reference
BNNS-OH	PA-6	10 wt%	0.499	[20]
Lys@BNNPs	PVA hydrogel	11.3 wt%	0.91	[21]
BNNS-Trp	EP	30 wt%	2.1	[23]
CBN	PC	20 wt%	0.7341	[28]
BNNS	TPU	30 wt%	5.15	[31]
BNNS@PDA	ANF	50 wt%	3.33	[33]
h-BN	BECy	15 vol%	0.55	[35]
3D-BN network	EP	34 vol%	4.42	[36]
BNNS	PPA	44 wt%	2.89	[37]
SPI-BNNS	TA@CNF	1 wt%	6.881	[38]
Lys@BNNSs	TPU	5 wt%	0.52	this work

## Data Availability

Data presented in this study are available on request from the corresponding author.

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
