# Peer review of "Amino Acid-Assisted Sand-Milling Exfoliation of Boron Nitride Nanosheets for High Thermally Conductive Thermoplastic Polyurethane Composites"

_polymers, 2022, doi:10.3390/polym14214674_

Round 1

Reviewer 1 Report

This paper presents valuable contribution in the field providing new and simple method for modification of HBN and its application to improve performance of TPU composites and therefore can be recommended for publication with few comments and suggestions for improvement

1.The authors should provide details what exactly equipment is used with the name and supplier. If conventional ball milling equipment is used with zirconia ball it is a ball-milling process not sending as this term is used and could confuse readers. Sanding means that sand particles are used and equipment is different.

2.Genarl comment about characterization section there are many missing info about conditions used for characterization of materials which should be included. For example is TGA done in air or nitrogen?

3. For TGA measurements it is important in include DTG graphs and determine Tmax value. In opinion of the reviewer more valuable results could be obtained using TGA in air that will provide better decomposition and potentially to observe two peak one for O and one for N. Authors should provide TGA/DTG graphs of control Lysine to confirm origin of these peaks.

4. Could authors provide elemental % composition from XPS and contest of N provide another evidence  about % of crafted lys in correlation with TGA results

Reviewer 2 Report

The paper by Zheng et al. deals with the Amino acid assisted sand-milling exfoliation of boron nitride  nanosheets for high thermally conductive thermoplastic polyu-3 rethane composites. This work is a very good contribution to the field and could be published in Polymers after minor revision as mentioned below:

1.     All abbreviation should be defined the first time used in the title and the manuscript

2.     CAS number, purities and providers of all chemicals should be added in the experimental section.

3.     English should be checked and improved. The paper contains a lot of grammatical and typographical errors.

4.     Relative error should be added to all values and Figures used in the manuscript (and just some of them). This error should determine the number of digits used after the decimal point.

5.     The obtained results should be compared to literature; A comparison table should be added.

6.     Some recent literature in this field should be added to the paper and discussed; see for instance: Nanomaterials, 2018, 8, 716 and Nanoscale, 2015, 7, 613-618
